# Increasing opal productivity in the Late Eocene Southern Ocean: Evidence for increased carbon export preceding the Eocene-Oligocene glaciation

Volkan Özen[1,2], Johan Renaudie[2], David B. Lazarus[2], Gabrielle Rodrigues de Faria[1,2]

[1]Freie Universität Berlin, Institute for Geological Sciences, Malteserstraße 74–100, 12249 Berlin, Germany
[2]Museum für Naturkunde, Leibniz Institute for Evolution and Biodiversity Science,
Invalidenstraße 43, 10115 Berlin, Germany

*Correspondence to*: Volkan Özen (volkan.oezen@fu-berlin.de)

**Abstract.** The Eocene/Oligocene Transition represents a period of profound changes in diatom productivity and evolutionary history within the Cenozoic era. Unraveling how these changes correlate with climatic shifts during this transition is crucial for understanding the potential role of diatoms in the cooling trends observed at the Eocene/Oligocene boundary (~33.9 Ma). Current research predominantly relies on bulk opal accumulation measurements to assess productivity dynamics, which fails to distinguish the contribution of different biosiliceous (e.g., diatom versus radiolarian) plankton to total biogenic silica productivity. Furthermore, despite the fundamental role of community composition and diversity in diatom productivity and carbon sequestration, these factors are often not incorporated in existing studies focusing on the late Paleogene diatom productivity. The main objective of our work is to explore the potential roles of diatom communities in the Late Eocene climatic changes by focusing on diatom- and radiolarian-specific productivity across multiple Southern Ocean sites, rather than bulk opal measurements, and by incorporating total diatom abundance into the analysis of diatom diversity evolution throughout the Eocene/Oligocene transition. By quantifying diatom and radiolarian abundances across four Southern Ocean sites in the Atlantic and Indian Ocean sectors, and analyzing diatom productivity through recent reconstructions of diatom diversity from approximately 40-30 Ma interval, our findings reveal a significant increase in diatom abundance coupled with notable shifts in community diversity. These changes suggest a potential ecological shift, likely associated with the development of stronger circum-Antarctic currents in the Late Eocene. Such shifts could have influenced the efficiency of the biological carbon pump by enhancing organic carbon export to the deep ocean and thus potentially contributing to reduced atmospheric $CO_2$ levels. While our findings indicate that the expansion of diatoms may have been a part of the mechanisms underlying the Late Eocene cooling, they also highlight the importance of integrating diatom diversity and community evolution into diatom productivity research. Furthermore, our results offer valuable insights into the complex relationship between diatom abundance and diversity in the geological record, reflecting the intricate interplay of environmental and climatic factors.

# 1 Introduction

The Eocene/Oligocene boundary (E/O, ~33.9 Ma) marks the end of the Cenozoic Hothouse with high-latitude surface ocean cooling and an abrupt 1.5 per mil increase in global benthic $\delta^{18}O$ values (Shackleton and Kennett, 1975; Zachos et al., 1996, 2001; Coxall et al., 2005; Zachos and Kump, 2005; Coxall and Pearson, 2007; Liu et al., 2009; Westerhold et al., 2020; Hutchinson et al., 2021). It corresponds to the largest cooling shift of the late Paleogene gradual cooling trend and the abrupt emplacement/expansion of the Antarctic ice sheet (Lear et al., 2008). Despite the extensive research, the underlying mechanisms are under dispute. The discussions on the possible mechanisms have revolved around three main domains (1) gradual thermal isolation of Antarctica with the development of the Antarctic Circumpolar Current (ACC) initiated by the deepening of the Southern Ocean (here and after SO) gateways (Kennett, 1977; Barker, 2001), (2) the threshold response of the Earth climate to the atmospheric $CO_2$ decrease in the late Paleogene (DeConto and Pollard, 2003; Ladant et al., 2014), and (3) the evolution of the west Antarctic rift system, which might have significantly modulated ice-sheet volume and climate feedbacks (Wilson and Luyendyk, 2009; Wilson et al., 2013).

These mechanisms are possibly interlinked; oceanographic change, atmospheric $CO_2$ drawdown, and tectonic reorganization of Antarctic topography are supported and extensively discussed based on proxy records and model results (e.g., Scher and Martin, 2006; Ladant et al., 2014; Elsworth et al., 2017; Paxman et al., 2019; Toumoulin et al., 2020; Hutchinson et al., 2021; Lauretano et al., 2021; Klages et al., 2024). Within this broader framework, several studies have suggested that increased SO productivity may have contributed to $CO_2$ decline by linking changes in circulation to export productivity and carbon sequestration (e.g., Diester-Haass and Zahn, 1996; Salamy and Zachos, 1999; Schumacher and Lazarus, 2004; Egan et al., 2013; Rodrigues de Faria et al., 2024). Although the timing and characteristics of this productivity shift remain debated (Renaudie, 2016; Wade et al., 2020; Bryłka et al., 2024,; Rodrigues De Faria et al., 2024), it was likely a piece of the broader mechanistic mosaic leading to the E/O transition and Antarctic glaciation. Diatoms and radiolarians, as major siliceous plankton groups, are pivotal to these discussions, both as contributors to export production and as proxies for changing nutrient supply and ocean circulation. Their fossil records suggest significant reorganization across the Late Eocene to Early Oligocene interval (see Section 1.2).

## 1.1 Opal as a paleoproductivity proxy

Biogenic silica deposition in modern oceans reflects surface ocean productivity, a pattern observed consistently across diverse regions from the equatorial to high-latitude Pacific and Atlantic Oceans (Baldauf and Barron, 1990; Barron et al., 2015). Biogenic silica has higher preservation potential than organic carbon (Tréguer et al., 1995; Ragueneau et al., 2000 and references therein), signifying the potential of biogenic silica deposition in tracking the changing paleoproductivity trends throughout the Cenozoic. However, this proxy is complicated by variability in silica dissolution and preservation, as well as the decoupling of the silica-carbon relationship from surface waters to sediments (Ragueneau et al., 2000; Abrantes et al.,

2016). The factors influencing silica dissolution and preservation are not fully constrained and are expected to vary significantly under different oceanographic and climatic conditions throughout the Cenozoic (Ragueneau et al., 2000; Westacott et al., 2021). Although the links between the opal deposition and productivity dynamics are complex, it has been shown that the secular trend of opal deposition is closely related to the global oceanographic and climatic changes (Cortese et al., 2004). Additionally, the evolutionary history of biosiliceous plankton underlying opal deposition during the Cenozoic is a critical but often overlooked aspect in paleoproductivity interpretations based on opal accumulation. Most available data are based on bulk opal measurements, which can obscure the contributions of different biosiliceous plankton groups, such as radiolarians (another extremely important siliceous plankton) and marine diatoms. Assessing the relative contributions of different biosiliceous plankton groups to opal sedimentation is essential for accurate paleoproductivity reconstructions (Ragueneau et al., 2000, 2006).

## 1.2 Prior opal records and the role of diatom diversity in Southern Ocean productivity

Evidence from the SO suggests that productivity in the region increased during the Late Eocene, with the first notable shifts occurring around 38–37 Ma (Diester-Haass and Zahn, 1996; Schumacher and Lazarus, 2004; Villa et al., 2014; Rodrigues de Faria et al., 2024). The rise in opal deposition during the Late Eocene and at the E/O (Salamy and Zachos, 1999; Diekmann et al., 2004; Anderson and Delaney, 2005; Bryłka et al., 2024), has been linked to the growing dominance of diatoms in open ocean settings. Given the central role of modern diatoms in carbon export through the biological carbon pump (e.g., Tréguer et al., 2018), these observations have drawn attention to a possible link between increased diatom productivity and atmospheric $CO_2$ decline at the E/O (Salamy and Zachos, 1999; Scher and Martin, 2006; Rabosky and Sorhannus, 2009; Egan et al., 2013; Renaudie, 2016). However, interpreting opal deposition history in terms of diatom productivity across the latest Eocene remains challenging because the data so far is based on bulk opal measurements which do not allow to assess the relative contribution of diatoms and other siliceous plankton, especially radiolarians. Although diatoms dominate opal sedimentation in modern oceans, radiolarians were more common in the early Paleogene, and the shift to diatom dominance occurred during a poorly constrained interval in the Middle to Late Paleogene (Renaudie, 2016).

The role of diatom diversity in driving productivity is also poorly constrained and often overlooked in paleoproductivity studies. Most reconstructions addressing Late Paleogene opal deposition do not distinguish among siliceous groups and remain agnostic to the species diversity underlying the opal signal (Salamy and Zachos, 1999; Diekmann et al., 2004; Anderson and Delaney, 2005; Plancq et al., 2014). Yet in modern ecosystems, diverse plankton communities are associated with higher biomass production, carbon export, and greater ecological stability (Tréguer et al., 2018; Virta et al., 2019; Hatton et al., 2024). A positive, often unimodal, relationship between diversity and productivity has been documented across many taxa, particularly groups of plants on global scales (Mittelbach et al., 2001). However, in paleoceanography, the link between diatom diversity and abundance/productivity is not well understood. Diatom diversity has at times been used as a proxy for abundance (Lazarus et al., 2014), though molecular data complicate this assumption by showing comparable

diversity values in both eutrophic and oligotrophic settings (Malviya et al., 2016). These findings point to the need to re-
evaluate the diversity-productivity relationship in fossil plankton communities using direct, paired observations.

In this study, we aim to improve constraints on SO diatom productivity across the Late Eocene–Early Oligocene interval by distinguishing the relative contributions of diatoms and radiolarians to total biogenic silica deposition. We present newly generated mass accumulation rate (MAR) data for both groups, based on the same sediment samples used in recent biological barium (bio-Ba) reconstructions (Rodrigues de Faria et al., 2024). By comparing these group-specific accumulation records with bulk opal and bio-Ba productivity estimates, we provide an independent assessment of siliceous plankton dynamics across the transition. Finally, we explore the relationship between diatom abundance and diversity to discuss the long-presumed link between diatom diversity and abundance.

## 2 Methods

### 2.1 Material

We generated diatom and radiolarian abundance data from samples collected from the following SO sites: Deep Sea Drilling Project (DSDP) Site 511 (Falkland Plateau, 51°00.28'S; 46°58.30'W) (sampled interval ~27–180 meter below sea floor (mbsf)), Ocean Drilling Project (ODP) Site 1090B (Agulhas Ridge, 42°54.8′S 8°53.9′E) (~188-335 mbsf), ODP Site 748B (Kerguelen Plateau, 58°26.45′S; 78°58.89′E) (~96–171 mbsf), and ODP Site 689D (Maud Rise, 64°31′S 3°6′E) (~104–132 mbsf) (Fig. 1). Our study examines 53 samples spanning the temporal interval from the Late Eocene to the Early Oligocene, approximately between 39 and 30 Ma, with site-specific coverage varying due to differences in sedimentation history at each site.

DSDP Site 511 (Falkland Plateau) and ODP Site 1090 (Agulhas Ridge) contain hiatuses in the earliest Oligocene, limiting the temporal coverage of these sites. At DSDP 511, our samples' temporal coverage is ~37.5–32.5 Ma. Similarly, at Site 1090, the sampled interval spans ~38–33 Ma, with a hiatus restricting samples from extending well into the Oligocene. In contrast, at ODP Site 689 (Maud Rise) and ODP Site 748 (Kerguelen Plateau), sedimentation is relatively continuous, providing a more complete record of the E/O transition (~36.5–30 Ma at Site 689; ~40–29 Ma at Site 748). All analyzed samples and corresponding measurements, including diatom and radiolarian abundance data, are detailed in the Supplementary Materials.

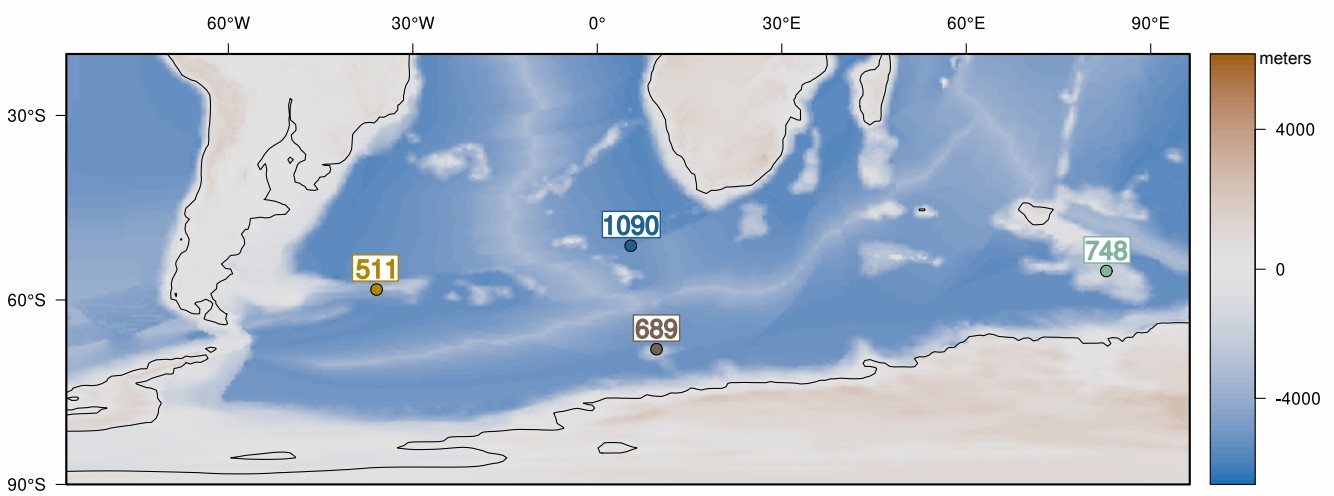

**Figure 1: Locations of the studied Southern Ocean sites, shown on a Late Eocene paleogeographic and paleobathymetric reconstruction (adapted from Straume et al. 2024). Estimated paleodepths at the E/O boundary: 2436 mbsl at DSDP Site 511; 2036 mbsl at ODP Hole 689D; 1270 mbsl at ODP Hole 748b and 3358 mbsl at ODP Hole 1090B (see Methods)**

In the southern high latitudes, DSDP Site 511 and ODP Site 1090 are notable for being a major locus of biogenic silica deposition across the E/O transition (e.g., Diekmann et al., 2004; Anderson and Delaney, 2005; Renaudie, 2016; Wade et al., 2020). While ODP Site 689D and 748B, in general, do not exhibit the same level of opal productivity, significant productivity changes have been documented at these sites across the E/O transition (e.g., Salamy and Zachos, 1999; Bryłka et al., 2024). These findings signify that these sites provide invaluable insights into the SO productivity in the areas proximal

to the Antarctic continent. They provide essential insights into how opal productivity varies under different regional settings, offering a broader perspective on productivity changes across the SO.

A comprehensive overview of the updated age models used in this study for each Hole/Site is available in Rodrigues de Faria et al. (2024); see also Supplementary Text 1. The models can also be accessed via the Neptune Database (Renaudie et al.,

2020, 2023). Paleobathymetry at each site at the E/O boundary was computed using those age models and each hole's lithological descriptions taken from their corresponding Initial Reports (Shipboard Scientific Party 1983, 1988, 1989, 1999) using PyBacktrack (Müller et al. 2018). The files used as input for PyBackTrack can be found in the Supplementary Materials, as well as its output.

### 2.1.1 Sample Preparation

Microscope slides for counting diatom and radiolarian abundances were prepared following a modified version of the methods described by Moore (1973) and Lazarus (1994) and sieved using a 10 µm sieve. About 0.5–1 gram sediment was treated with hydrogen peroxide ($H_2O_2$) and pentasodium triphosphate ($Na_5P_3O_{10}$) over heat, followed by treatment with hydrochloric acid (HCl). The resulting solution was then sieved through a 10 µm sieve. A controlled amount of the residues

was gently settled over three coverslips at the bottom of a beaker. This approach ensures the material settles randomly across the coverslips, minimizing potential biases that might arise during the enumeration phase (for details, see Lazarus, 1994).

## 2.2 Diatom and radiolarian absolute abundance and accumulation rates

Absolute abundances (ab) for diatoms and radiolarians were calculated by counting specimens on a known area of slides, following the equation below:

$$ab = N \times (Ab/Am) \times (Vp/Vu) \times 1/w \tag{1}$$

with N is being the number of specimens counted, Ab the area of used beaker (6079 mm2), Am the area measured in mm$^2$, Vp the volume of residue prepared in mL, Vu the volume used in mL and w the weight of the dry sediment in gram.

Accumulation rates of diatoms and radiolarians were calculated by multiplying abundance values with the shipboard measured dry bulk densities and the linear sedimentation rates (LSR). The LSR values used are based on the updated age models for the targeted sites.

## 2.3 Diatom abundance and diversity

Our study also examines the relationship between diatom diversity and total diatom abundance, which is essential for understanding the influence of diversity on overall community productivity in diatoms. Rather than relying on bulk opal accumulation rates, which do not distinguish the relative contribution of different siliceous groups like radiolarians, we focused on diatom-specific abundance values. This approach provides a clearer understanding of the relationship between diversity and abundance within diatom communities.

To explore these interactions, we compared recent diatom diversity reconstructions (Özen et al., subm.) with diatom abundance data obtained in this study across the E/O transition. This approach allowed us to directly examine how variations in diatom diversity correspond to changes in abundance and to explore the potential implications of these interactions for overall diatom productivity. In our comparisons, by focusing on diatom abundances per gram of sediment, we aimed to minimize potential biases associated with accumulation rates, which can be affected by uncertainties in age models. This approach ensures a more accurate representation of the abundance-diversity relationship, offering valuable insights into the ecological and environmental factors that influenced diatom productivity during the E/O transition.

## 3 Results

Our diatom MARs reveal a clear latitudinal organization in the Late Eocene SO.This is expected, as diatoms are a major contributor of biogenic sedimentation at sub-Antarctic sites, DSDP Site 511 (Falkland Plateau) and ODP site 1090 (Agulhas Ridge), where biogenic silica is the main sedimentary component across the study interval (Renaudie, 2016; Wade et al., 2020; see Fig. S1).

In the sub-Antarctic Atlantic,ODP Site 1090 diatom MAR increased from ~38 Ma onward, peaking around 36.8 Ma, closely matching previously published bulk opal MARs (Diekmann et al., 2004; Anderson and Delaney, 2005, see Fig. 2d and 2e). Site 1090 had an average diatom MAR of $1.26 \times 10^7$ frustules cm$^{-2}$ kyr$^{-1}$ (standard deviation (std. dev.) = $9.48 \times 10^6$; range: $5.88 \times 10^5$ to $3.26 \times 10^7$; total number of samples, N = 15). At DSDP Site 511, diatom MARs gradually rose throughout the Late Eocene, peaking near 33.4 Ma ($1.76 \times 10^8$, see Fig. 2e), with notably higher accumulation rates (mean = $5.73 \times 10^7 \pm 4.97 \times 10^7$; range: $1.65 \times 10^7 - 1.77 \times 10^8$, N = 9). The overall diatom MAR trends at this site align well with recent bulk opal accumulation rates (Bryłka et al., 2024, Fig. 2d). Fig. 3a shows the distributional characteristics of diatom MARs at both sites.

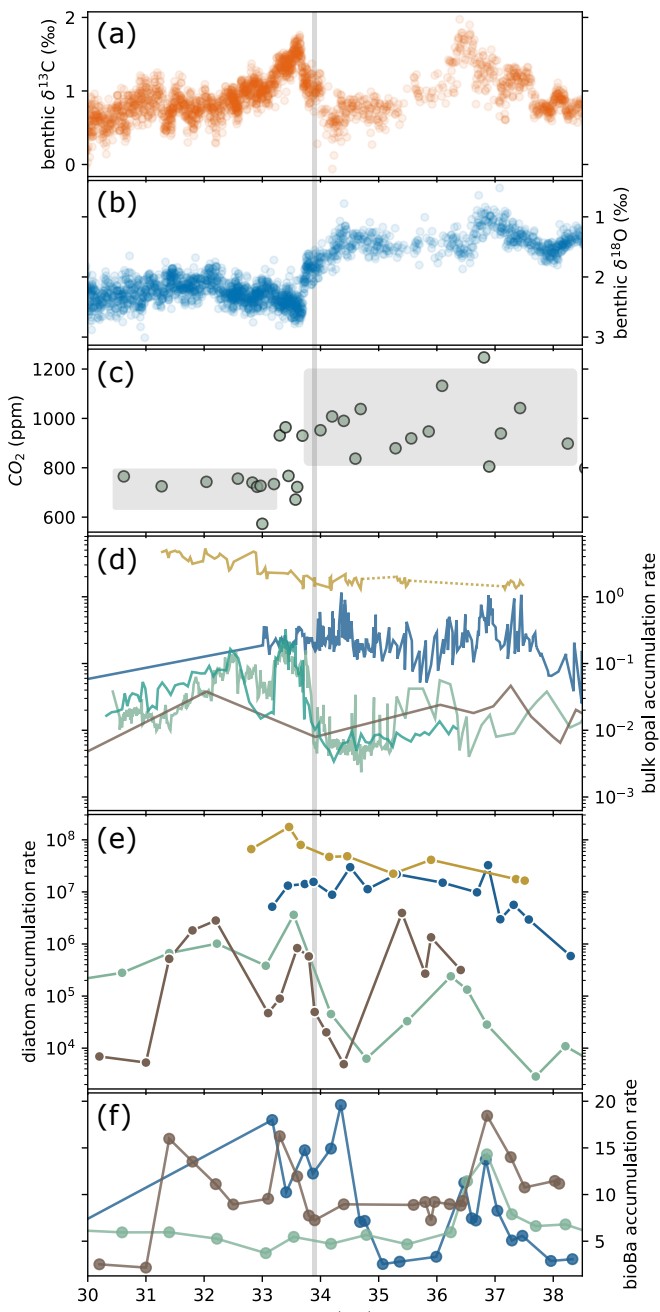

Figure 2. (a) Global composite benthic foraminiferal $\delta^{13}C$ and (b) $\delta^{18}O$ records (from Westerhold et al., 2020). (c) $CO_2$ compilation (from Zhang et al., 2013; Anagnostou et al., 2020). (d) Bulk opal accumulation rates (gr cm$^{-2}$ kyr$^{-1}$, solid lines) from DSDP 511 (yellow, Bryłka et al., 2024), ODP 1090 (blue, Diekmann et al., 2004; Anderson and Delaney 2005), Kerguelen Plateau ODP Sites (light green, 744 and 738; dark green 748) (compiled from Ehrmann, 1991; Ehrmann and Mackensen, 1992; Salamy and Zachos, 1999, Bryłka et al., 2024), and ODP 689 (Faul and Delaney, 2010). (e) Diatom mass accumulation rates (MARs) (diatom cm$^{-2}$ kyr$^{-1}$; scatter points with solid lines; this study) and (f) biogenic barium (bioBa) accumulation rates (µmol cm$^{-2}$ kyr$^{-1}$) from ODP Sites 1090, 689, and 748 (from Rodrigues de Faria et al., 2024).

The Antarctic sites showed lower and more variable diatom MARs (Fig. 2e and 3a). At ODP Site 689 (Maud Rise), diatom

MAR exhibited prominent peaks (within the temporal precision of our data) at ~35.5 Ma, 33.6 Ma, and 32.3 Ma, with an

average value of $7.92 \times 10^5$ frustules cm$^{-2}$ kyr$^{-1}$ (std. dev. is $1.16 \times 10^6$, range: $4.92 \times 10^3 - 3.94 \times 10^6$, N = 16; see Fig. 2e). At

ODP Site 748 (Kerguelen Plateau), diatom and bulk opal MARs are in agreement after ~37.5 Ma, suggesting increasing

diatom contribution to the total opal productivity towards the E/O boundary. Mean diatom MAR at this site is $3.74 \times 10^5$,

with minimum and maximum values of $2.86 \times 10^3$ and $3.63 \times 10^6$, respectively (N = 18). At this site, it has been shown that

across the Middle Eocene, other siliceous groups, ebridians and radiolarians, dominate the record (Witkowski et al., 2012).

Combined with our results, this suggest that diatom dominance in the Kerguelen Plateau region started towards 37 Ma,

which is consistent with our sample surveys as there is a strong presence of ebridians in our samples preceding ~38 Ma (See

Supplementary Data 1). Moreover, our results show that, compared to the other sites, diatom MAR at ODP Site 748B

changed substantially between the Eocene and Oligocene (Fig. 3a), with a mean of $4.8 \times 10^4$ frustules cm$^{-2}$ kyr$^{-1}$ in the

Eocene and 1.03 in the Oligocene.

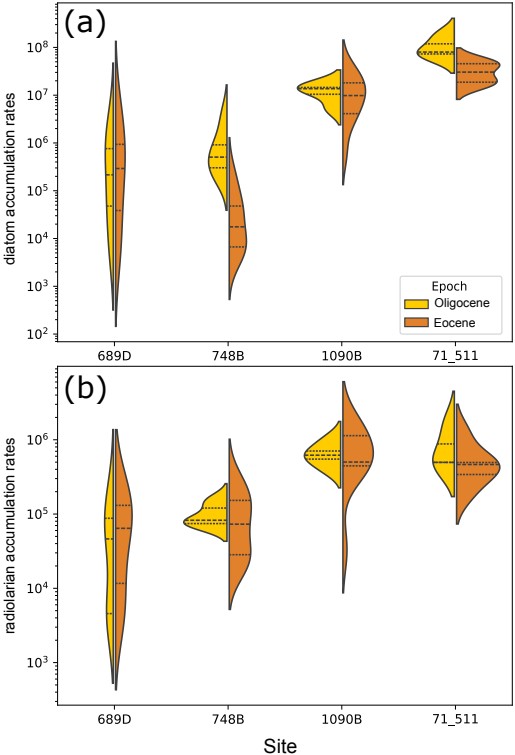

**Figure 3. Distribution of MARs (specimen cm$^{-2}$ kyr$^{-1}$; in log scale) for (a) diatoms and (b) radiolarians at each studied site (x-axis). Eocene and Oligocene samples are shown separately (see legend). Lines within each distribution indicate the quartiles, marking the median, the 25th and 75th percentiles.**

Our diatom MARs, combined with published bulk opal MAR records, reveal two distinct intervals during the E/O transition, 36.5–35.5 Ma and 34–33 Ma, when opal productivity at Antarctic sites (ODP 689 and 748) sharply increased, approaching sub-Antarctic Atlantic levels (Fig. 2d and 2e). Although sub-Antarctic sites (DSDP 511 and ODP 1090) maintained consistently higher opal productivity throughout the E/O transition, Antarctic sites experienced transient but significant increase in diatom MARs during these intervals, bringing opal flux levels closer to those observed at sub-Antarctic sites.

Radiolarian MAR patterns are broadly in agreement with diatom MARs across the E/O transition, derived from the same samples (Fig. 4c). At ODP Site 1090, radiolarian MARs showed two prominent peaks at ~37 and 34.5 Ma. The mean accumulation rate at this site is $7.43 \times 10^5$ radiolaria cm$^{-2}$ kyr$^{-1}$ ($\pm 4.56 \times 10^5$, range: $3.29 \times 10^4$–$1.60 \times 10^6$). At DSDP Site 511, MAR values remained relatively stable throughout the Late Eocene, exhibiting significant increases towards and after the E/O boundary, at approximately 34 Ma and 33.5 Ma (mean = $6.34 \times 10^5 \pm 4.53 \times 10^5$, range: $1.79 \times 10^5$–$1.56 \times 10^6$).

At ODP Site 748, radiolarian MAR peaked notably at ~36.5 Ma, followed by a substantial decline, and recovered to pre-E/O levels only around 30.5 Ma (mean = $1.01 \times 10^5 \pm 7.61 \times 10^4$, range: $1.73 \times 10^4$–$3.06 \times 10^5$). ODP Site 689 indicated two prominent peaks, between 36–35 Ma, and at ~31.5 Ma, with a lower average MAR of $6.49 \times 10^4$ ($\pm 6.20 \times 10^4$, range: $3.89 \times 10^3$–$1.85 \times 10^5$). From ~35 Ma onward, radiolarian MAR differences between sub-Antarctic Atlantic (sites 1090 and 511) and Antarctic (sites 689 and 748) sites became more pronounced, reflecting an increasing contrast in accumulation rates during the latest Eocene–Early Oligocene (Fig. 4c; see also Fig. S2).

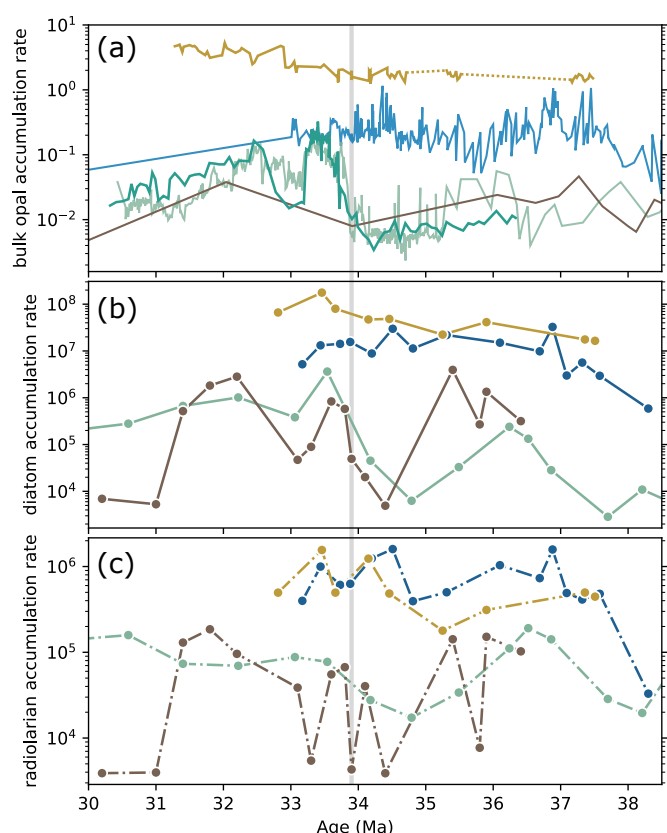

**Figure 4.** Comparison of (a) bulk opal, (b) diatom, and (c) radiolarian accumulation rates (specimen cm⁻² kyr⁻¹) at SO sites. Site colors are consistent across figures. Diatom and radiolarian data are from this study, while bulk opal accumulation rates (a; g cm⁻² kyr⁻¹, solid lines) are compiled from the following sources: DSDP 511 (yellow; Bryłka et al., 2024), ODP 1090 (blue; Diekmann et al., 2004; Anderson and Delaney, 2005), Kerguelen Plateau sites ODP 744 and 738 (light green) and ODP 748 (dark green) (Ehrmann, 1991; Ehrmann and Mackensen, 1992; Salamy and Zachos, 1999; Bryłka et al., 2024), and ODP 689 (Faul and Delaney, 2010).

### 3.1 Correlations between diatom diversity and abundance

Diatom abundance and diversity showed varied correlations among sites across the Late Eocene–Early Oligocene interval (Fig. S3). At DSDP Site 511, exhibiting the highest species diversity, diatom abundance and diversity were in great agreement across the E/O transition (Fig. 4c; see also Fig. S3). At ODP Site 1090, diatom abundance and diversity were generally synchronous, except between approximately 37 and 34.5 Ma, during which diversity values stayed relatively low and constant while abundance values showed the highest values. Moreover, the pronounced peaks in diatom abundance around 36.8 and 34.5 Ma did not correspond with similar increase or trend shift in diatom diversity (Fig. 5d). In contrast, at ODP Sites 689 and 748, diatom diversity and abundance values were in agreement (Fig. 5b and 5e ; Fig. S3). At ODP Site 748, there was a substantial increase in both bulk opal (Bryłka et al., 2024, Fig. 5a) and diatom abundance, reaching values

similar to those seen at sub-Antarctic Atlantic sites ODP 1090 and DSDP 511. However, diversity values remained significantly lower compared to those documented at the sub-Antarctic Atlantic sites (Fig. 5e).


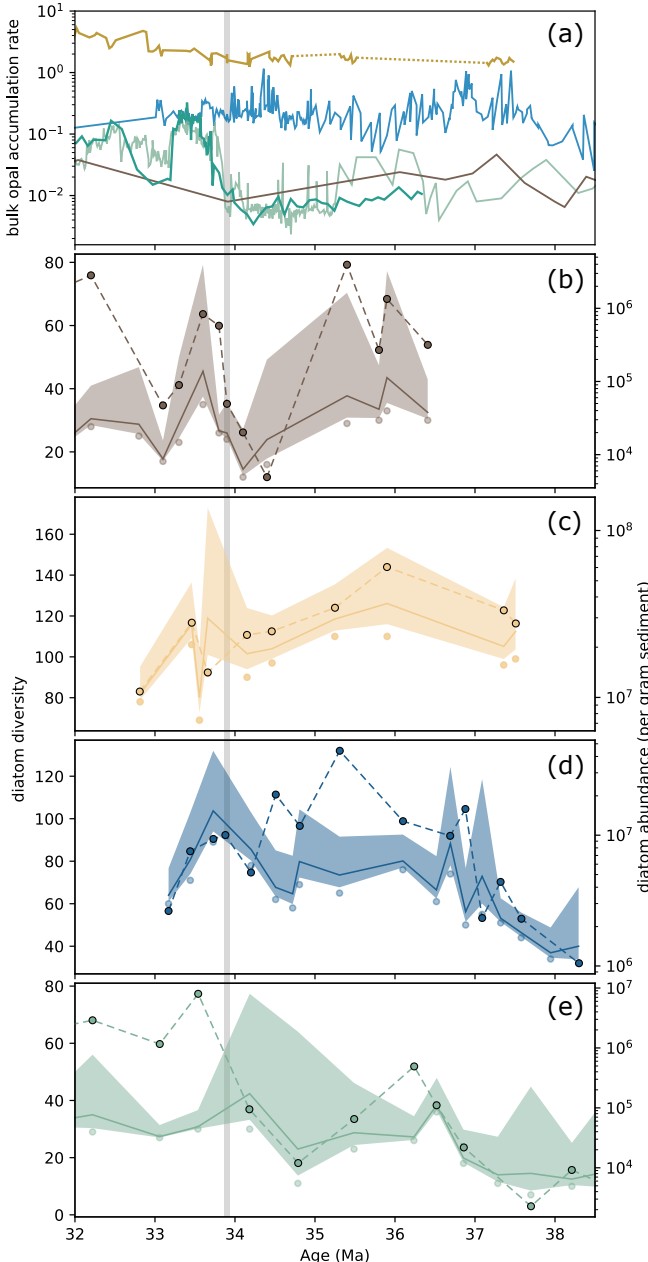

**Figure 5. Comparison of bulk opal accumulation rates (gr cm⁻² kyr⁻¹) with diatom diversity (number of species) and abundance. (a) from DSDP 511 (yellow, Bryłka et al., 2024), ODP 1090 (blue, Diekmann et al., 2004; Anderson and Delaney, 2005), Kerguelen**
**Plateau ODP Sites (light green, 744 and 738; dark green 748) (compiled from Ehrmann 1991; Ehrmann and Mackensen 1992;**

Salamy and Zachos, 1999; Bryłka et al., 2024), and ODP 689 (brown, Faul and Delaney, 2010) (b-e) Diatom diversity (scatter points) with Chao1 diversity estimates (solid line, 95% confidence envelope) (Özen et al., subm.) and diatom abundance per gram sediment (scatter points, dashed lines).

## 4 Discussion

**4.1 Diatom and radiolarian productivity and opal record across the Middle-to-Late Eocene transition (~36–38 Ma)**

Although the timing and mode of opal productivity (that is, the relative contribution of diatoms and radiolarians, the underlying diatom diversity, and whether the flux is pulsed or gradual) differ among sites, our results suggest a substantial reorganization of SO diatom and radiolarian productivity between ~38 and 36 Ma. In the sub-Antarctic Atlantic, bulk opal at ODP Site 1090 shows a gradual increase from 38 Ma onwards, intensifying by ~37 Ma (Diekmann et al., 2004; Anderson

and Delaney, 2005). Our diatom MARs closely follow this trend, suggesting growing diatom dominance in the region. This shift was not confined to a single site or proxy but occurred across multiple sectors: radiolarian communities in the South Pacific reorganized (Pascher et al., 2015), and in the Indian Ocean sector both bulk opal and diatom MARs rise between ~38-36 Ma (Fig. 2d-e, ODP 748 and 744), in parallel with evidence for increasing eutrophic conditions from calcareous nannofossil**s** (Villa et al., 2014, see Fig. S4), confirming that the diatom signal reflects a real plankton productivity shift.


In the Atlantic sector, benthic foraminiferal accumulation rates (Diester-Haass and Zahn, 1996, see Fig. S5) and radiolarian communities (Funakawa and Nishi, 2008) record this shift in productivity and environmental conditions. Bio-Ba records from ODP Sites 1090, 689, and 748 (Rodrigues de Faria et al., 2024) document a ~37 Ma export productivity maximum that is synchronous with the diatom MAR rise at ODP Site 1090 but leads peaks at Antarctic-adjacent sites (ODP 689, 748) by

~0.5–1 Myr (Fig. 2e). This offset does not reflect differences in age models, as identical samples and age models were used in both datasets, and likely reflects regional environmental controls during the Middle-to-Late Eocene, including latitudinal differences in sea-surface temperature (e.g., Douglas et al., 2014; Sauermilch et al., 2021), variations in nutrient distribution, and circulation patterns influenced by still-shallow SO gateways (e.g., Sauermilch et al., 2021; Rodrigues de Faria et al., 2024).


What does this productivity reorganization across the Middle-to-Late Eocene transition (~38–36 Ma) signify? It encompasses a suspected ephemeral East Antarctic glaciation, namely the Priabonian Oxygen Maximum (~37 Ma; Scher et al., 2014). This event is marked by a sharp negative Neodymium (εNd) excursion within a broader Late Eocene positive trend (Scher and Martin, 2006; Scher et al., 2014; Wright et al., 2018). Previous studies link Priabonian Oxygen Maximum

cooling to productivity shifts across SO sectors (e.g., Villa et al., 2014; Pascher et al., 2015; Rodrigues de Faria et al., 2024) through mechanisms involving transient intensification and organization of a proto–Antarctic Circumpolar Current (proto-ACC) that enhanced frontal upwelling and nutrient delivery (e.g., Rodrigues de Faria et al., 2024). This interpretation is consistent with modelling results suggesting that even a shallow opening of the Drake Passage, which likely occurred during

the late Middle-Eocene (Scher and Martin, 2004; 2006), could have reorganized ocean flow and promoted proto-ACC formation (Toumoulin et al., 2020). A comparable mechanism operates in the modern SO, where the strength of the latitudinal temperature gradient controls westerly wind intensity, which governs ACC transport and the intensity of wind-driven upwelling (Rintoul et al., 2001). In parallel, Priabonian Oxygen Maximum cooling, in line with southern high-latitude sea-surface temperature compilations (O'Brian et al., 2020), would have steepened the temperature gradient, intensified the westerlies that drove the proto-ACC, and increased Ekman divergence, delivering nutrient-rich waters to the surface ocean. Silicon isotope data further support this scenario, indicating increased diatom silicic acid utilization during this interval and pointing to enhanced silicic acid supply to the surface ocean via intensified upwelling (Egan et al., 2013). Consistent with this, although site level responses vary, diatom and radiolarian MARs between 38 and 36 Ma show positive covariation at the Agulhas Ridge and the Kerguelen Plateau. This pattern is more consistent with a shared physical driver, enhanced upwelling and nutrient supply, than with competitive replacement under constant nutrient conditions.

We note that, in addition to diatoms and radiolarians, other sources of biogenic silica, such as sponge spicules, silicoflagellates and ebridians, can also contribute to bulk opal, which may complicate direct comparisons with group-specific records. In our samples, these groups are not a significant component. Our focus therefore remains on diatoms and radiolarians to assess how their contributions changed through time within the broader biogenic silica pool, rather than attempting a one-to-one correspondence with bulk opal records.

Viewed in broader context, these productivity changes in the diatom and radiolarian records and community composition across the Middle-to-Late Eocene transition (~36–38 Ma) (e.g., Pascher et al., 2015; Özen et al., subm.) are interpreted here as a response to the intensification of SO circulation, likely reflecting stronger eastward flow and enhanced vertical exchange within the developing proto-ACC system, and the associated increase in nutrient distribution and upwelling. At the same time, this transition marks the onset of global rise in diatom abundance and diversity (Renaudie et al., 2016), and in our records diatom MARs show a net increase at all sites except ODP 689 during the subsequent interval, pointing to a basin-wide reorganization of diatom export. Comparable opal productivity surges during ~38–36 Ma are recorded in the equatorial Atlantic (Nilsen et al., 2003; Fig. S6) and northern Atlantic (Witkowski et al., 2021), suggesting that reorganization had a broad geographic reach and may reflect the strengthening of a cross-latitudinal circulation system akin to the modern Atlantic Meridional Ocean Circulation (AMOC). Indeed, it has been proposed that from ~38 Ma onwards this circulation started to strengthen under the effect of increasing circum-Antarctic circulation, the proto-ACC, which is an integral part of the cross-latitudinal circulation across the Atlantic (Borrelli et al., 2014). This aligns with our diatom accumulation rates, which show a substantial reorganization 38 Ma onwards, aligning with earlier paleoproductivity reconstructions suggesting a substantial productivity increase across the SO sites between 36–38 Ma (e.g., Diester-Haass and Zahn, 1996; Pascher et al., 2015; Rodrigues de Faria et al., 2024). Although the precise timing and sequence of SO gateway opening and the development of circum-Antarctic circulation patterns remain debated (Diester-Haass and Zahn, 1996; Mackensen, 2004;

Stickley et al., 2004; Scher and Martin, 2006; Livermore et al., 2007; Barker et al., 2007; Hodel et al., 2021; Evangelinos et al., 2024), the balance of evidence suggests a Late Eocene strengthening of circum-Antarctic circulation, possibly during the Middle-to-Late-Eocene transition, that set the stage for a large-scale reorganization of SO productivity and the growing dominance of diatoms.

## 4.2 Opal pulse across the Eocene/Oligocene boundary

Two overarching patterns characterize SO productivity across the E/O boundary (~35.5–32 Ma): (1) strong regional heterogeneity (see Bryłka et al., 2024; Rodrigues De Faria et al., 2024), and (2) distinct latitudinal responses. During the latest Eocene, diatom and radiolarian MARs diverge between sub-Antarctic (DSDP 511; ODP 1090) and Antarctic sites (ODP 689; ODP 748). At the Antarctic sites, both groups decline (Fig 4b-c), consistent with low bulk-opal values on the Kerguelen Plateau and bio-Ba signals (Fig. 2d and 2f). In contrast, sub-Antarctic records show high latest-Eocene productivity: bulk opal, diatom MARs, and bio-Ba peak near ~34.5 Ma, especially at Agulhas Ridge (ODP 1090), while the Falkland Plateau (DSDP 511) maintains high diatom productivity (see Fig 2d-e).

Latitudinal divergence between sub-Antarctic and Antarctic sites strengthens from ~35.5 Ma onward (Fig. 4a-c; Fig. S2). This divergence is also evident in radiolarian productivity: before ~35.5 Ma, radiolarian MARs co-vary between the two regions, but from 35.5 Ma onward they diverge, signalling a change in biogeography and productivity. Indeed, radiolarian endemism in the southern high latitudes rises from ~35.5 Ma (Lazarus et al., 2008), consistent with greater regional isolation or reorganization of water masses. Tectonic reconstructions point to further Tasmanian Gateway deepening at about the same time (Stickley et al., 2004), although Nd-isotope data imply that fully developed deep throughflow likely did not establish until the Neogene (Evangelinos et al., 2022). A step-like increase in the negative Ce anomaly at ~35.5 Ma indicates increased oxygenation of thermocline and bottom waters in the SW Pacific (Hodel et al., 2022), and the authors link this change to Tasmanian Gateway tectonic evolution and enhanced vertical mixing. Against this background circulation change, the Kerguelen Plateau region records a gradual ecological transition from a radiolarian-dominated to a diatom-dominated phase (Fig S7). This shift occurs while overall opal flux remains low in the Antarctic-adjacent sites, pointing to a change in the biological source of silica, with production increasingly driven by diatoms. Such a transition implies a more direct coupling between silica utilization and organic carbon export, even under relatively low total fluxes. Broader confirmation of circulation reorganization comes from dinocyst biogeography and sedimentological evidence, which record stronger SO circulation and surface cooling from ~35.7 Ma (Houben et al., 2019).

Taken together, these lines of evidence indicate that circum-Antarctic circulation, which had already begun to strengthen across the Middle-to-Late Eocene transition, underwent further intensification from ~35.5 Ma. We interpret this reinforcement of circulation and vertical mixing as the main driver of the growing divergence between sub-Antarctic and Antarctic sites. Model simulations are consistent with this view: experiments with Late-Eocene boundary conditions show

that progressive gateway deepening enhanced eastward circumpolar flow, reorganized upper-ocean circulation, and shifted deep-convection zones northward toward ~40 ºS, encompassing the Agulhas Ridge region (Toumoulin et al., 2020). This circulation shift has indeed been linked to substantial export productivity at ODP Site 1090 in the latest Eocene (Rodrigues De Faria et al., 2024; Fig. 2f), which possibly also underlies the opal productivity burst and overall high diatom productivity we observe at this site (Fig. 2d-e). At the same time, this northward shift of deep-convection would have reduced circulation strength in Antarctic-proximal sectors, particularly the Weddell region (see Toumoulin et al., 2020), which in turn would have reduced upwelling, nutrient supply, and export production (Rodrigues De Faria et al., 2024). The combined effect of a further strengthening proto-ACC and a weakened Antarctic circulation system offers a plausible mechanism for the sustained decline in diatom and radiolarian productivity at Antarctic sites from ~35.5 Ma to the E/O boundary.

In Antarctic-adjacent sites, the latest Eocene low-productivity regime shifts at the E/O boundary. Diatom MARs rise sharply, closely matching bulk-opal accumulation (Fig 2d) and coinciding with the largest increase in global $\delta^{18}O$ values (Fig. 2b), suggesting a link between the East Antarctic glaciation, cooling, and enhanced diatom productivity. Clay-assemblage studies suggest stronger physical weathering in the earliest Oligocene at Maud Rise (ODP 689) and the Kerguelen Plateau (ODP 748) (Robert et al., 2002). Such weathering likely increased silica input and fueled higher productivity in these regions. Consistently, the earliest-Oligocene Nd-isotope excursion in the Kerguelen Plateau, tied to glaciation and weathering (Scher et al., 2011), strongly correlates with opal flux (see Fig. S8), reinforcing the link between continental discharge and silica supply. In contrast, radiolarian productivity does not return to early Late-Eocene levels (Fig. 4c). Instead, it remains low while diatoms increase strongly, suggesting that diatoms progressively gained dominance, likely reflecting their competitive advantage in utilizing the available silicic acid.

We note that geographically variable diatom flux across SO sites may not necessarily imply regionally inconsistent forcing. Sub-Antarctic sites supported diverse diatom communities (see Section 4.3) and already sustained high fluxes in the Late Eocene, likely operating close to ecological carrying capacity, which may have muted the magnitude of their response. The biological basis for such a ceiling is well captured by the relationship between abundance and silicic acid in coastal upwelling zones: sedimentary diatom abundance increases as silicic acid concentrations rise until a threshold is reached, beyond which further silicic acid input yields little additional diatom accumulation (Abrantes et al., 2016). This diminishing return complicates efforts to trace a coherent sequence of diatom productivity and oceanographic reorganization across the SO, because an increased silica supply through upwelling in already highly productive sub-Antarctic regions, such as DSDP 511 and ODP 1090, may have altered community composition or frustule silicification rather than producing a proportional increase in diatom productivity and thus opal flux.

### 4.3 Diatom diversity and productivity: A cause/effect relation?

One of the defining features of SO opal productivity across the E/O transition is its parallel with major changes in both diatom and radiolarian community composition (Funakawa and Nishi, 2008; Lazarus et al., 2008; Pascher et al., 2015; Özen et al., subm.). Under the influence of the dynamic climatic and oceanographic features of the SO, and within the limits of the fossil data resolution, it is a complex task tracking the exact ecological response dynamics of the biosiliceous plankton. Recent diversity reconstructions (Özen et al., subm.) show that Late Eocene–Early Oligocene SO diatom assemblages were at least five times more diverse than previously documented, revealing a pronounced increase in species richness and a highly dynamic community composition throughout this interval. Such diversity is expected to be positively associated with a broader range of functional traits within the community (Tréguer et al., 2018), and therefore with more efficient nutrient utilization, which is one of the operating terms of the biological carbon pump (Farmer et al., 2021).

The close relationship between observed diversity and diatom MARs and abundance, however, does not necessarily reflect a simple cause-effect link. Observed diversity could increase as a function of abundance through ecological interactions, or it could be an artefact of higher opal flux, better frustule preservation, and thus more morphologies recorded. Our findings suggest that while there is a notable correlation between diversity and abundance (see Fig. 5b-e), the relationship is not straightforward, reflecting a more intricate interplay between these two metrics. At ODP Site 748, for instance, diatom abundance rises sharply in the earliest Oligocene (Fig. 5e), yet diversity remains relatively low, barely reaching 30 species. This contrasts with observations at ODP Site 1090, where similar abundance values are associated with much higher diversity, suggesting that abundance alone does not drive diversity (Fig. 5d). Furthermore, at ODP Site 1090, a period of consistently high diatom abundance between 36.5 and 34.5 Ma corresponds to relatively low and stable diversity, indicating community stability rather than a direct abundance-diversity coupling. Notably, this interval at ODP Site 1090 is marked by the dominance of a specific diatom genus, *Pyxilla*, (Özen et al., subm.) which likely contributed to the observed stability in diversity despite high overall abundance. Interestingly, in the second opal pulse at ODP 1090, diversity declines while abundance remains high, further illustrating that diversity and abundance need not covary, nor can diversity be reduced to a function of preservation alone.

On the other hand, data from DSDP Site 511 reveal a strong alignment between diatom abundance and diversity (Fig. 5c). This can be interpreted in an ecologic context, suggesting that the strength of the diversity-abundance relationship can vary considerably depending on the site-specific conditions, community composition, and the associated functional groups. Comparable patterns are seen in the modern ocean: metabarcoding surveys and sediment trap records indicate that diatom diversity is not uniformly coupled to abundance but instead reflects the balance between a few dominant species and many rare ones (e.g., Malviya et al., 2016; Rigual-Hernández et al., 2019), likely structured by regional circulation and ecological filtering. This perspective reinforces the importance of incorporating biological and ecological dimensions into

palaoproductivity studies on diatoms, in line with previous work emphasizing the role of community composition in maintaining ecosystem function and the efficiency of carbon export through diatomaceous pathways (Tréguer et al., 2018). Taken together, our results suggest that the interplay between diatom diversity and abundance is not merely additive, but a feedback loop modulated by external environmental conditions.

## 4.4 Diatom productivity and its possible role in the E/O cooling

Studies to date indicate that, across the E/O transition, SO regions experienced substantial shifts in productivity (e.g., Diester-Haass and Zahn, 1996; Diekmann et al., 2004; Anderson and Delaney, 2005; Egan et al., 2013; Villa et al., 2014; Plancq et al., 2014; Pascher et al., 2015; Rodrigues de Faria et al., 2024). These shifts form the basis for the hypothesis that increasing productivity contributed to $CO_2$ drawdown through the biological carbon pump (Salamy and Zachos, 1999; Scher and Martin, 2006; Egan et al., 2013). Our results add weight to this view: records across the E/O transition show rising

diatom accumulation together with increasing community diversity (Özen et al., subm.), consistent with Si isotope evidence for pulses of silica utilization and associated changes in carbon export (e.g., Egan et al., 2013). We emphasize, however, that diatom productivity was not an overriding mechanisms in itself but one element within a broader climatic and oceanographic mosaic that together shaped atmospheric $CO_2$ drawdown across the E/O boundary.

A frequent criticism of a diatom-driven increase in productivity and its potential role in E/O cooling is that opal-rich sediments are restricted to a few regions, such as the Agulhas Ridge and Falkland Plateau regions (e.g., Wade et al., 2020). However, this view largely reflects sediment classification systems that emphasize the most abundant component and have historically favored carbonate-rich deposits. Several biases contribute to the underrepresentation of biogenic silica in the deep-sea sediment record: (1) a bias towards carbonate pelagic sedimentation due to substantial carbonate rock weathering

on land, (2) pelagic primary sediment names based on the most abundant single sedimentary component, and (3) a historical preference in deep-sea drilling for well-preserved carbonate sections, often chosen for geochemical studies. As a result, compilations which rely only on dominant sediment types, mostly reflect pelagic carbonates while underestimating the presence of biogenic silica. In contrast, studies using quantitative estimates of biogenic silica (e.g., smear slide analyses, Renaudie, 2016) provide a more accurate, though still incomplete, picture of silica accumulation. Given these limitations, the

absence of opal-dominated sediments in broad sediment classifications does not contradict the evidence for increased biogenic opal deposition. Indeed, our results show a clear rise in biogenic opal across all targeted sites, providing robust evidence for enhanced diatom productivity across the E/O transition.

However, the mode and magnitude of the opal deposition/preservation vary between sites, reflecting local depositional

settings and preservation conditions. Despite these differential geological filters, intervals of productivity reorganization identified in earlier studies are also evident in our records. Importantly, these productivity events coincide with changes in

dominant species composition and community structure (Özen et al., subm.), which may have tuned the efficiency of the biological carbon pump (Tréguer et al., 2018).

Diatoms are particularly effective exporters of organic carbon (Ragueneau et al., 2000; Tréguer et al., 2018). Diatom-dominated export buffers particulate organic carbon (POM) against microbial decomposition in the mesopelagic zone far more effectively than coccolithophore-dominated fluxes (e.g., Cabrera-Brufau et al., 2021). Thus, the Late Eocene increase in diatom abundance and diversity would have improved the efficiency of the biological carbon pump, enhancing atmospheric $CO_2$ drawdown even without a marked rise in total SO productivity. This effect gains further weight considering

that the Eocene $CO_2$ levels were already close to threshold values (~750 ppm) thought necessary for Antarctic ice sheet initiation (DeConto and Pollard, 2003), though we note that such thresholds are model-dependent and vary with boundary conditions (Gasson et al., 2014). Thus, increasing oceanic productivity and the greater efficiency of diatom-mediated carbon export may have provided the final touch that pushed $CO_2$ levels below the boundary conditions, contributing to the E/O climate shift.

**5 Conclusion**

This study focuses on the dynamics of diatom productivity and community diversity across the Late Eocene, as the Cenozoic Hothouse came to an end and the SO underwent a major productivity reorganization from ~38 Ma onward. Our findings show a marked increase in diatom abundance alongside major reorganizations in both diatom (Özen et al., subm.) and radiolarian (Lazarus et al., 2008; Pascher et al., 2015) communities, consistent with an overall rise in SO productivity (e.g.,

Diester-Haass and Zahn, 1996; Anderson and Delaney, 2005; Villa et al., 2014; Rodrigues de Faria et al., 2024). These changes point to a broad evolutionary and productivity shift, likely driven by major changes in ocean circulation, including the early development of the AMOC (e.g., Borrelli et al., 2014) and the strengthening of the circum-Antarctic currents (e.g., Houben et al., 2019; Sarkar et al., 2019; Rodrigues de Faria et al., 2024).

Bulk opal records are not fully informative for understanding diatom productivity and its impact on the biological carbon pump and atmospheric $CO_2$. This is because bulk opal measurements (1) fail to differentiate between contributions from diatoms and other biosiliceous plankton, such as radiolarians and (2) overlook the biological background of productivity, particularly diversity and community composition, which is a critical component of diatom-mediated carbon sequestration (Tréguer et al., 2018). Our study integrates both the diversity and abundance dynamics of diatoms across the E/O transition,

revealing that increases in both likely enhanced the efficiency of the biological carbon pump from 38 Ma onward. This enhancement reflects not only higher diatom abundance but also more effective nutrient utilization linked to diversity, which together may have supported a stronger carbon flux to the ocean interior for a sustained period of time. We therefore highlight SO diatom expansion, through both abundance and diversity, as an important component of the Late Eocene

carbon cycling. While the precise strength of the link between diatoms and global cooling across the E/O transition remains uncertain, our results support the view that diatoms contributed to the efficiency of the biological carbon pump during this critical interval, and that their role deserves continued attention in understanding the mechanisms behind Cenozoic climate dynamics.

## Data availability

The supplementary information and raw data are available in the Supplement, and the raw data will be available upon publication on Zenodo (10.5281/zenodo.14826336, Özen et al., 2025).

## Supplement

The supplement related to this article is available online at:

## Author Contribution

VÖ collected and analyzed the data and drafted the manuscript. All authors contributed to revision and editing of the final version.

## Competing interests

The contact author has declared that none of the authors has any competing interests.

## Acknowledgements

We thank Sylvia Dietze (MfN Berlin) for her assistance with sample preparation.

## Financial Support

This study was funded by the Federal Ministry of Education and Research (BMBF) under the "Make our Planet Great Again – German Research Initiative", grant number 57429681, implemented by the German Academic Exchange Service (DAAD).

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
