# Peer review of "Increasing opal productivity in the Late Eocene Southern Ocean: Evidence for increased carbon export preceding the Eocene-Oligocene glaciation"

_EGUsphere, 2025_

## Author Response (AR1)

**Dear Editor and Reviewers,**

We are grateful for your constructive and insightful comments. We have addressed all suggested revisions, and we hope that the revised manuscript now meets your expectations.

Please find our responses below.

All the best,

Volkan and co-authors

**Reviewer #1**

When talking about bulk opal measurements, authors predominantly refer only to diatoms and radiolarians. On occasion, sponge spicules often comprise a large proportion of biogenic silica in Eocene sediments. Considering their overall much larger size, even smaller abundance could contribute to equal BSi value. Further silicoflagellates, ebridians, and archaeomonds can reach higher concentrations as well. I think it's worth including it, either in the introduction or the discussion in case bulk opal measurements cannot be explained only by diatoms and radiolarians (with the current representations of the graphs it is difficult for the reader to determine).

We thank you for this note. These groups, as other important sources of biogenic silica, are not significantly present in our samples, which is why we did not include them in our analyses. We agree, however, that acknowledging their potential role improves the clarity and transparency of our paper, and we have added a short paragraph on this point in the Discussion (Section 4.1, Lines 316-320)

Fig. 2.d. It is confusing and difficult to correlate when bulk opal accumulation and diatom accumulation are overlapping and on different scales. I would suggest changing it to the same scale or presenting it on separate graphs, as I can imagine the diatom ranges would completely flatten on the same scale. Potentially making a graph where diatoms and radiolarians are on the same scale and bulk opal accumulation rates are e.g., above would be easier and clearer to analyze.

Thank you for pointing this out. We have revised Figure 2 by separating the bulk opal and diatom results into two subplots, as suggested.

Fig. 2. I would suggest making a line or shaded area across the graph to mark the E/O. It would be easier to read the results.

Done. The E/O boundary is now marked with a line in each plot.

Fig. 3. As mentioned above radiolarian accumulation rates could be shown in one figure with diatoms. It is difficult to compare across figures. Also please be consistent with figure

labeling and presentation. In Figure 2 diatom accumulation rate on the graph is written without the unit, in Figure 3 radiolarian accumulation rate is written with the unit.

Thank you for this helpful suggestion. We have revised the figures accordingly. Radiolarian, diatom, and bulk opal accumulation rates are now combined within a single figure as separate subplots, making direct comparison easier. Please note that the radiolarian results are now shown in Figure 4 (instead of Figure 3). For consistency, we standardized the axis labeling by moving the units into the figure captions. In addition, radiolarian data are replotted on a logarithmic scale to make them directly comparable with the diatom and bulk opal datasets.

Fig. 4. Likewise I suggest marking the E/O across the graph.

Done.

In line 418 authors write "...further questioning the assumption that these two metrics, diversity and abundance, are directly linked, or that observed diversity is primarily controlled by preservation." I indeed agree, that diversity is not necessarily controlled by abundance, or as authors point it is a complex feedback loop. However, preservation is important to consider. It's often easy to observe whether assemblage is well/poorly preserved when observing how the frustule presents itself on light and scanning electron microscope. Lack of lightly silicified genera, which are characteristic of Eocene such as Asteropmhalus or small genera could indicate higher dissolution rates. I think overall it is always worth making general notes on the preservation state. Potentially authors can go back to slides and make general observations about whether high diversity, high abundance samples have better frustule preservation.

Thank you for this comment. We fully agree that preservation is a key factor in evaluating the relationship between diatom diversity and abundance. In response, we have added our preservation assessments for each sample. These are now presented in Figure S1 and detailed in the Appendix of the Supplementary Figure file. In our study, we classified samples by examining frustule conditions of lightly silicified groups, dissolution features, and frustule fractionation patterns.

Was the sieved fraction (below 10 um) checked for the diatoms? Diatoms in genera eg. Actinoptychus, Paralia can be as small as 6-8 um and still easily distinguished under the light microscope. Overall I doubt omitting this fraction caused a large bias in the data, however, sieved fraction when looking at the abundances should always be checked.

We occasionally checked the sieved fraction during the sample preparation and did not observe a significant number of diatoms being excluded. We therefore concluded during the data collection phase omitting this fraction does not bias our abundance data.

My biggest comment would be about how the authors calculated absolute abundances. The formula used by authors is indeed used broadly in the literature, especially for paleo studies. Another method, which in my subjective opinion, is more accurate and less biased for such calculations is divinylbenzene (DVB) microspheres (Battarbee and Kneen, 1982). In these calculations, only dry sediment weight and microsphere concentrations are needed,

which would introduce fewer potential errors than the authors' calculations. It might be beneficial to look at a few samples (e.g. 5) and use microspheres to establish concentrations and compare the results. This method is more used in freshwater studies on diatoms, however, it is a good chance to introduce it to the paleo world as well.

Thank you very much for this methodological suggestion. Implementing this method was not feasible within the scope of this study, as our research group has since been dismantled and relocated from the Museum für Naturkunde Berlin, where the main data preparation and collection were carried out, and we no longer have access to the facilities there. We are grateful for this recommendation and will certainly consider applying the approach in our future research.

Line eg. 51, 76, 80: missing comma in the citation. Please check the whole manuscript for this type of correction.

Done. Thank you for pointing these out. We have corrected the citations and reviewed the manuscript to address typos and other minor errors.

In line 52, the authors write "...poorly constrained timing of the SO gateway...", the reader can gather SO corresponds to the Southern Ocean however for clarity in line 49, the authors should write "...focuses particularly on the deepening of Southern Ocean (here and after SO) gateways."

It is now revised and clarified in Line 42, which reads: '...deepening of the **Southern Ocean (here and after SO)** gateways...'

Line 54: ca (circa) please correct to ca.

We have removed this part from the introduction section, as other reviewers noted that the section was too long.

Line 67: Please correct CO2 to $CO_2$.

Done.

Line 235: Be consistent with the usage of hyphens and en dash. For age ranges always use en dash. Please check and correct where necessary through the text.

Done.

Lines 238–240: Please avoid one-sentence paragraphs. Try to incorporate in previous.

Done. We incorporated it into the previous paragraph.

**Reviewer #2**

Thank you for giving me the opportunity to review this manuscript. As pointed out by the referee 1, the manuscript is overall well written and pleasant to read. It focusses on new diatoms and radiolarian data as evidence for increased carbon export preceding the Eocene Oligocene transition. Overall, I recommend this manuscript for publication after minor to major revisions. I might sound a bit harsh in the comments below, but I did enjoy reading this manuscript, and I think it is worthy of publication after revisions. As you'll see, most of my comments are minor.

We do thank you for your constructive comments.

Overall, although nicely written, the manuscript is quite long, and I would suggest to try to shorten it by being more concise.

Thank you. We have revised the manuscript to improve clarity and reduce length. In addition to addressing specific comments, we removed redundant phrasing. We hope the revised text now reads more concisely.

Abstract: I feel like you talk a lot about radiolaria here but maybe under-used them in the main text?

We agree and have revised the manuscript to expand the discussion of radiolarian results in the main text. In addition, we added two new figures illustrating different aspects of the radiolarian data (Fig. 3 and Fig. S2) and expanded the figure representing the radiolarian results (now Fig. 4)

L36: "ocean cooling" maybe add "by X °C"?

Thank you for this suggestion. We have chosen not to include a specific temperature estimate, as the magnitude of surface cooling across the E/O boundary remains debated and varies considerably among regional proxy data. For example, Liu et al., (2009) reported ~6 °C drop, whereas more recent work (Tibett et al., 2023) does not support such a dramatic change. Cooling at the boundary is well established, but the degree remains controversial. We therefore consider a qualitative statement more appropriate.

Liu, Z., Pagani, M., Zinniker, D., DeConto, R., Huber, M., Brinkhuis, H., Shah, S. R., Leckie, R. M., and Pearson, A.: Global Cooling During the Eocene-Oligocene Climate Transition, Science, 323, 1187–1190, https://doi.org/10.1126/science.1166368, 2009.

Tibbett, E. J., Burls, N. J., Hutchinson, D. K., and Feakins, S. J.: Proxy-Model Comparison for the Eocene-Oligocene Transition in Southern High Latitudes, Paleoceanog and Paleoclimatol, 38, e2022PA004496, https://doi.org/10.1029/2022PA004496, 2023.

L39: missing dot after reference.

Done. Thank you for pointing this out –several references with missing dot, and we have now corrected them all.

L98 to L102: this has already been developed in the introduction earlier, avoid repetition. Overall the introduction is quite (too) long in my opinion.

We have significantly revised the text to remove repetitions and have also shortened the Introduction.

L102 and throughout the text: the reference to Özen et al., submitted is not accepted and not available to read. Lots of sentences rely on this reference which is yet to be accepted, so I am not sure about the validity of using it yet.

The diversity values we cite are based on our recent work, which is currently in its second round revision, and hopefully will be accepted soon. The reviewers of that manuscript recommended publication after revision, and their comments focused on structure and language (they also noted that the text was too long, a point raised here as well, which gives a consistent message to the lead author) rather than the integrity of the data and analyses. To ensure transparency, we now provide the full manuscript as a preprint (see DOI: 10.31223/X50N1B)

Methods: please add the latitudes and longitudes of investigated core sites, especially as you investigated records of various locations, it is important to quickly find this information. If possible also add the paleolocation?

We have added the latitude and longitude coordinates for each site in the Methods section (2.1 Material). Figure 1 has also been revised to display the site paleo-coordinates on the paleogeographic map, allowing quicker reference to the paleolocations of the studied sites.

Regarding this, it would be nice to have latitudes and longitudes available on figure 1 too. Maybe this figure can be use to show current opal export too for example, instead of just depth?

Done. We have revised the map to include latitude and longitude. However we did not add the current opal export as we thought adding modern data on a paleogeographic map was confusing at best.

L175: is "ab" the absolute abundance?

Done. We added "Absolute abundance (ab)"

L190: again reference not available

Please see our response to your comment on L102.

L191: Capital T after dot

Done.

L198: Maybe define MAR?

Done.

Fig2: maybe connect the points too for panel C? Like in D and E?

Thank you for this suggestion. We chose not to connect the data points in panel C, as our focus is on the broader state shift indicated by the available $CO_2$ data rather than on individual point-to-point trends. This shift is illustrated by the gray-shaded areas, which highlight the distinct ranges of Eocene and Oligocene $CO_2$ values.

FIG 2 Not sure what to suggest but panel D contains lots of graphs and is not easy to read.

We have revised Figure 2 by separating the bulk opal and diatom data into two subplots to make the results easier to follow.

L216: use of "fluctuate" is a bit strange

We have revised the text, which now reads: 'Antarctic sites showed lower and more **variable** diatom MARs (Fig. 2e and 3a).'

L228-229: last sentence sounds a bit strange

We have revised it (Line 220 in the revised text): "These Antarctic productivity peaks temporally **brought opal flux levels closer to those observed at sub-Antarctic sites**."

L232: Taking age models into account, and so on, I'm not sure about "coincided", as we have a 0.5 to 1 ma gap while we could expect a faster / synchronous response?

To avoid implying synchronicity, we have replaced '*coincided*' with '*broadly aligns*' in the revised text. Please see our response to the comment below.

L235: "corresponds" Maybe use "is concomitant" or something similar? To me, "corresponds" induces a causal effect.

We have revised the text to use more cautions language and avoid implying precise synchronicity/causality. It now reads (Lines 222-227)

"These intervals, of Antarctic and sub-Antarctic diatom MAR convergences, unfolded under distinct climatic conditions. The first (~36.5–35.5 Ma) interval **broadly aligns** with the late Eocene warming event, **which has been documented** at multiple high-latitude SO sites (ODP Site 689 (Maud Rise); ODP Sites 738, 744, 748 (Kerguelen Plateau); DSDP Site 277; Diester-Haass and Zahn, 1996; Bohaty and Zachos, 2003; Villa et al., 2008, 2014; Pascher et al., 2015). In contrast, the later interval (~34-33 Ma) **broadly concomitant** with a sharp increase in global foraminiferal $\delta^{18}O$ in the earliest Oligocene, signaling substantial cooling and the onset of permanent Antarctic glaciation (Fig. 2b and 2d)."

L238-240: just a few more words on that?

We have incorporated this short paragraph into the previous one for better flow and to avoid redundancy.

L242: So radiolarian responded 0.5 earlier than diatoms? Of course taking sample resolution into account.

We would like to clarify that both diatom and radiolarian measurements are based on exactly the same set of samples and age models. Any apparent offset in timing therefore does not reflect differences in sampling resolution, but rather variability in the response dynamics of the two groups and/or differences in preservation. To make this clear, have we revised the Results section to note that the same samples were used for both groups.

Please see Lines 231-232: "Radiolarian MAR patterns are broadly in agreement with diatom MARs acro ss the E/O transition, **derived from the same samples (Fig 4c)**".

L231-236: So one peak corresponds to a warming and the other one to a cooling?

When compared with previous studies and observations, data suggest that the two convergence intervals occurred under contrasting conditions. The first (~36.5–35.5 Ma) coincides with a late Eocene warming event documented across several high-latitude Southern Ocean sites, whereas the later interval (~34–33 Ma) aligns with a pronounced $\delta^{18}O$ increase marking substantial cooling and the onset of Antarctic glaciation. We discuss this pattern in detail in Sections 4.1 and 4.2.

L252: "This reorganization": remove "is"

Done.

L255 and throughout this paragraph and in the manuscript too: One of the biggest issues of this manuscript for me. Although not critical: I feel like I'm guided to see what the authors want me to see because the trends in their records is not so obvious. I can see want they say for the bioBa, but not always for the diatom records that are new here, and looking at figure 2, I can see two diatom records that hardly vary, and two that vary a lot, one of which stay high after the transistion, and the other one that increase, decrease, and so on, so the trend is not very clear here in my opinion. I think this should be discussed more and try to not oversell what we can actually see.

Thank you for this comment. We understand the concern that some of the trends in the diatom records appear less obvious compared to the bio-Ba data. This difference arises because our diatom MARs are plotted on logarithmic scale, whereas the bio-Ba records are shown on a linear scale. We chose a logarithmic scale for diatom values to better represent the large spread in accumulation rates across sites and to reduce the dominance of extreme values. To address potential confusion, please see the plot below, based on ODP 1090 data, showing the diatom records on a linear scale. This illustrate that, while the diatom patterns are not perfectly aligned with bio-Ba trends (as discussed in the text -Lines 289-295), both proxies capture the broad 'two-pulse' reorganization, providing a robust signal of productivity change.

[Figure]

Fig. 3 and in the text: Can we really talk about peaks here? I guess yes, but for diatoms, peaks are orders of magnitude higher than low values, while for radiolarian, it is "only" 2 or 3 times higher.

Thank you for this observation. We have revised the figure (now Fig. 4) by combining diatom and bulk opal data with radiolarian results to allow a clearer comparison, and we now show radiolarian results on a log scale for consistency. Regarding the use of the term 'peaks', please let us note that diatoms and radiolarians are expected to respond with different magnitude sto the same environmental shifts, as they operate in different biological (e.g., reproduction strategies, nutrient utilization, competing for available silicic acid), temporal (e.g., life cycles), and spatial (surface versus deeper water layers) domains. For this reason, radiolarian peaks are expressed as smaller relative changes than diatom peaks, but we still use the term to describe the intervals of increased accumulation rates within each group. The magnitude of change is therefore not directly comparable between the two groups, but both records show internally consistent peaks relative to their baseline values.

Fig. 3: Keep in mind that resolution is relatively low, so a bit difficult to draw conclusions like "peaks", it could also be isolated values. Any margin of error for this?

Please see our response above.

Maybe find a way to fuse Fig 2 and 3 for a direct comparison? Also maybe draw a line or something at the E/O transition?

We have revised the figure representing radiolarian results (previous Fig. 3, now Fig 4) and added diatom and bulk opal values to allow direct comparison. As suggested, we have also added a line in the plots to mark the E/O boundary.

Fig.4: Again, lots of data in this manuscript seems to rely on unpublished data, yet to be accepted and not available.

Please see our response to your comment on L102.

L284: add color after ODP689.

Done.

Fig.4: Maybe put the side captions on all panels?

Thank you for this suggestion. We chose to keep shared side captions to avoid repeating the same axis label in each subplot.

(discussion) Again I question the certainty with which conclusions can be drawn with this "Low" resolution. maybe try to be a bit more moderate?

We acknowledge the importance of being cautious when working with datasets of relatively low resolution. To address this, we have moderated the language throughout the manuscript. At the same time, the patterns based on diatom MARs are consistent with higher-resolution bulk opal records, indicating that our data provide a reliable approximation to higher resolution signals. This approximation suggest that our records capture the dominant productivity trends at the temporal scale most relevant to the questions addressed in this work.

-L299-303: I agree with this, but I am not sure all of this is truly visible in your data without being guided to see it.

Thank you for this critical comment. We acknowledge that some of the described patterns may not be immediately visually evident at first glance (please see our response with figure to your comment on Line 255). We have revised the text to present our observations more cautiously and descriptively. We also emphasize that our focus is on broad changes rather than the precise timing of individual shifts. In varying degrees and magnitudes, our data indicate broad reorganization across both the middle-to-late Eocene transition and the E/O boundary. The corresponding part your comment referring has been revised, please see Lines 324-330.

-L310: "environmental factors": such as?

We have revised the text. It now reads (Lines 291-295):

"This offset does not reflect differences in age models, as identical samples and age models were used in both datasets, and likely reflects **regional environmental controls** during middle-to-late Eocene, **including latitudinal differences in sea-surface temperature**

**(e.g., Douglas et al., 2014; Sauermilch et al., 2021), variations in nutrient distribution, and circulation patterns influenced by still-shallow SO gateways (e.g, Sauermilch et al., 2021; Rodrigues de Faria et al., 2024)**."

-L323: "circum-Atlantic": Atlantic? Antarctic?

Fixed.

-L337-340: More productivity is usually less dissolved O2 that is consumed, so why is it the opposite here?

We would like to clarify that the PrOM refers to the Priabonian Oxygen Maximum event (Scher et al., 2014), which is a benthic δ18O excursion (please see Figure 4 in Scher et al., 2014) interpreted as a possible indicator of ephemeral East Antarctic glaciation. Nd records from the same interval (Scher et al., 2014) further interpreted as associated weathering discharge and possible nutrient supply, which we discuss in our manuscript in the context of productivity changes.

L359: missing "is" between abundance and particularly?

Fixed.

L372: "a striking pattern" again, with this resolution, I would be more moderate. Plus, avoid this repetition at the end of paragraph.

Thank you. We have deleted the part referred to in this comment, as suggested by other reviewers to keep the Discussion section more streamlined. We have also moderated the wording throughout the text, as suggested. The pattern based on diatom MARs are consistent with higher-resolution bulk opal records, indicating that our data provide a good approximation to higher-resolution datasets and that resolution is not a major limitation.

L405: remove "simply"

Done.

L449-450: I don't understand this sentence, what are primary sediment names?

We have clarified the wording at (now) Lines 452–453, where 'primary sediment names' was intended to mean 'dominant sediment types.' The text has been revised accordingly. It now reads: "As a result, compilations which rely only on **dominant** sediment **types**, mostly reflect pelagic carbonates while underestimating the presence of biogenic silica."

L456-457: avoid long sentence with phrasing like "as a natural consequence of the fact that". I can be just replaced by "as". Overall try to be more concise, the manuscript is quite long as a natural consequence of the fact that there are lot of phrasing like this.

Thank you for this suggestion. We have revised and improved the text in the highlighted lines, replacing the phrasing as recommended. In addition, we shortened the manuscript by cutting redundant wording to make it more concise overall.

L470: I'm not sure what is "enhanced" here? Please rephrase.

As suggested, we revised it, and it now reads (Line 471): "Thus, increasing oceanic productivity and **the greater efficiency** of diatom-mediated carbon export …"

Last paragraph of the conclusion: It sounds a bit like: " This is what we propose for now until we find something more consistent to say" I now this is not what you mean so I recommend rephrasing it.

Thank you. We have revised the discussion section and removed the part in question from the conclusion.

**Reviewer #3**

The introduction is not focused on the problem the authors want to address in the paper. The introduction is too long, and while the numerous subsections are good, not everything is needed to discuss the authors' main focus: the quantification of diatom and radiolarian contribution to biogenic silica productivity during the E/O. I would recommend making it shorter and presenting only the essential information needed to discuss what the authors want to clarify, rather than creating a kind of review of the E/O boundary in Antarctica.

We agree with this comment. We have revised and shortened the introduction to keep the focus on the main objectives of the paper: quantifying the contributions of diatoms and radiolarians to biogenic silica productivity, and examining the relationship between diatom diversity and productivity. To this end, we removed the more detailed sections on the E/O boundary, Antarctic glaciation, and regional environmental evolution.

Material: Please provide a lithological column showing different lithologies and hiatuses well illustrated on the column. Then, please also briefly describe the key lithologies and major lithological changes, if there are any.

We have prepared stratigraphic column sections for each site and provided them as Supplementary Figure S1.

There is no description or explanation about the depth-age model used at each site. Because the authors discuss temporal disparities of diatom productivity, I request that the authors add a paragraph clearly explaining the age model used, along with its potential error margins.

We have now provided detailed descriptions of the age models used in our study, including the calculated error margins. These are available in the Supplementary Materials (**Supplementary Text 1**).

This file is also referenced in the main text at Line 139-140: "**A comprehensive overview of the updated age models used in this study for each Hole/Site is available in Rodrigues de Faria et al., 2024; see also Supplementary Text 1. The models can also be accessed via the Neptune Database (Renaudie et al., 2020, 2023)**"

Overall, I think the approach is good, but the authors do not consider silicoflagellates at all in the manuscript. For instance, in the Pacific, like the Japan Sea, they were also important primary producers during the Middle Miocene, similar to diatoms. So why not consider them as well? If they are not significant in the Southern Ocean, please state this; if they are significant but you did not assess them, please revise the manuscript by focusing specifically on the radiolarian and diatom contributions rather than claiming to cover all siliceous plankton.

Thank you for this important point. We have not intended to claim coverage of all siliceous plankton, but rather to focus on the contributions of diatoms and radiolarians. To clarify this point, we have stated in the Methods section that our main focus is these groups. Also, we have added a paragraph in the Discussion (Section 4.1) explicitly noting that other sources of biogenic silica, such as sponge spicules, silicoflagellates, and ebridians, can also contribute to bulk opal and may complicate direct comparisons with group-specific records. In our samples, however, these groups are not significant components. Our focus therefore remains on diatoms and radiolarians to evaluate how their contributions changed through time within the broader biogenic silica pool.

Please be more quantitative and not only describe the timing but also indicate average values, minimal and maximal values with the standard deviation, and the total number of samples (N).

Thank you for this suggestion. We have revised the Results section (Section 3, accordingly. To further illustrate our findings and their quantitative dimensions, we have also added a new figure that shows the distributional characteristics of the MAR values for both diatoms and radiolarians.

I am not sure that sections 3.1 and 3.2, which mix results with discussion, are needed here. I suggest going straight to the discussion to avoid redundancy.

We agree. Section 3.1 (Correlation of opal abundance to other paleoproductivity proxies) has been deleted. Section 3.2 (now Section 3.1; Correlations between diatom diversity and abundance) has been revised to present only results. We have retained this section as it provides a concise presentation of the 'diversity and abundance' results and associated figure (Fig. 5).

The discussion is overall very redundant, with similar topics being discussed in different paragraphs, making the end of the manuscript difficult to read. I don't think there are that many things that can be discussed with the new data proposed in this study. I suggest making only 2 parts: 4.1 "Diatom and Radiolarian Productivity vs. opal records" and 4.2 "Diatom diversity and Productivity: A cause/effect relationship." Then delete all the small subchapters that are very similar to each other.

Thank you. We have removed two sections from the Discussion and reorganized the text into four subsections, revising the text to keep it concise and repetition-free. The discussion of '*Diatom and Radiolarian Productivity vs. Opal Records*' is now structured into two subsections (Section 4.1 and 4.2), which address two critical transitional intervals: the middle-to-late Eocene transition (~38–36) and changes around the E/O boundary. The specific radiolarian subsection has been removed, and radiolarian text is now included into these two parts. We kept the section on diatom productivity and its link to E/O cooling (noting that we do not suggest diatom productivity as a sole/overriding mechanism underlying this climatic shift), as it is central to the broader implications of our results.

L.39: please add a dot after "Lear et al. 2008."

Done. We have also gone through the text and revised all citation formatting.

L.41-45: I think there is a third way, and it is the tectonics of Antarctica and the evolution of the Ross Sea rifting as well, which modulate Antarctic topography and thus affect the volume of ice-sheet able to be carried and therefore may affect climate changes. see:

Wilson, D. S., & Luyendyk, B. P. (2009). West Antarctic paleotopography estimated at the Eocene-Oligocene climate transition. Geophysical Research Letters, 36, 4 PP. https://doi.org/200910.1029/2009GL039297

Wilson, D. S., Pollard, D., DeConto, R. M., Jamieson, S. S. R., & Luyendyk, B. P. (2013). Initiation of the West Antarctic Ice Sheet and estimates of total Antarctic ice volume in the earliest Oligocene. Geophysical Research Letters, 40(16), 4305–4309. https://doi.org/10.1002/grl.50797

Thank you very much for this detailed suggestion. We have incorporated these points into the introductory text where we present the literature on the mechanistic background of the Antarctic glaciation at the E/O. Please see Lines 40-45 –the revised text is also below:

"The discussions on the possible mechanisms have revolved around three main domains (1) gradual thermal isolation of Antarctica with the development of the Antarctic Circumpolar Current (ACC) initiated by the deepening of the Southern Ocean (here and after SO) gateways (Kennett, 1977; Barker, 2001), (2) the threshold response of the Earth climate to the atmospheric CO2 decrease in the late Paleogene (DeConto and Pollard, 2003; Ladant et al., 2014), **and (3) the evolution of the west Antarctic rift system, which might have significantly modulated ice-sheet volume and climate feedbacks (Wilson and Luyendyk, 2009; Wilson et al., 2013)**."

L.59-60: Please explain why you think E/O pCO2 reconstructions are not well constrained and tell us what is the key factor hampering these reconstructions.

Thank you for this comment. To clarify, our intention in the text ("… while the CO2 reconstructions are accepted at face value, the possible mechanism(s) underlying the late Eocene drawdown in CO2 is not well constrained.") was not to suggest that CO2

reconstructions themselves are poorly constrained (although they are\*), but rather that the mechanisms driving the $CO_2$ drawdown remain uncertain. We have deleted this part to shorten the Introduction, as suggested, and we limited the text to listing and introducing these mechanisms rather than discussing their details.

\*Uncertainties in $CO_2$ reconstructions arise from underlying assumptions (as with any proxy), and especially for the Paleogene, the limited number of available data points. Although additional data exist across the EOT, we restricted our use of $CO_2$ records to two marine resources (Zhang et al., 2013; Anagnostou et al., 2020; please see Fig. 2c) to minimize biases associated with combining many different proxies (of $CO_2$ reconstruction). The uncertainty surrounding the late Paleogene $CO_2$ reconstructions is also illustrated in Hönisch et al. (2023, Figure 1). Let us note that this, however, does not change the fact that these $CO_2$ reconstructions are extremely valuable for understanding long-term climate evolution.

Hönisch, B., … , Zachos, J. C., and Zhang, L.: Toward a Cenozoic history of atmospheric $CO_2$, Science, 382, eadi5177, https://doi.org/10.1126/science.adi5177, 2023.

L.58-64: I understand the authors' meaning, but I would like them to explain and relate climatic dynamics and Southern Ocean oceanography in more detail. Which kinds of climatic events would enhance shallow water mixing? What type of topography would enhance current circulation? Why would pro-ACC strength enhance primary productivity? I think we have many questions in mind reading the current text, so I would suggest revising this part in more detail.

Thank you for these constructive comments. In line with other suggestions to shorten the Introduction, we have deleted this part to avoid a 'mini-review' style of the Introduction section. However, we have addressed these questions in more detail in the Discussion (Sections 4.1 and 4.2). There, we elaborate on Southern Ocean oceanographic evolution in relation to gateway tectonics/paleogeography, and on the links between proto-ACC development, shallow water mixing, and primary productivity (see revised text, also Lines 300-314)

L.67-72: I agree, but it is also very simplistic to attribute such a change in diatom productivity uniquely as well. This is why I would recommend carefully considering the tectonics and rifting of Antarctica with the reference I suggested, for highlighting the influence of geography on current paths and topography, which influence the potential volume of ice sheet that can be held by Antarctica and thus potentially impact pCO2. Rather than trying to explain that the E/O event is 100% caused by silica plankton blooms, I believe it is much fairer to propose clarifying the real contribution of silica plankton blooms to the E/O events among other events, including tectonic ones.

We agree that our original text gave the impression of attributing too much emphasis to diatom productivity. Our intention in the Introduction was to outline the basis for the hypothesized link between increased Southern Ocean productivity and E/O cooling. In line with one of your main comments, we have shortened the Introduction section and deleted this part to avoid a 'mini-review' of E/O changes. In the Discussion, we state that Southern

Ocean productivity, particularly the increasing role of diatoms, may have provided a '*final touch*' within the broader set of mechanisms driving the E/O transition, including oceanographic and tectonic factors, rather than representing an overriding or sole driver.

Lines 441-443 reads: "We emphasize, however, that **diatom productivity was not an overriding mechanisms in itself but one element within a broader climatic and oceanographic mosaic** that together shaped $CO_2$ drawdown across the E/O boundary."

Lines 472-473: "Thus, increasing oceanic productivity and the greater efficiency of **diatom-mediated carbon export may have provided the *final touch* that pushed** $CO_2$ levels below the boundary conditions, contributing to the E/O climate shift."

L.85-90: I think this paper's objectives should be related to what is highlighted here and not try to overestimate the role of diatoms during the E/O. Diatoms bloomed because a substantial change occurred or a threshold was exceeded in other factors controlling Antarctica, but it was not diatom rise that caused the big changes. I think they are one of the consequences of other factors much more related to paleogeography and topography.

Thank you. We have revised our text to make it explicit that we do not propose diatom productivity as a sole mechanism underlying E/O cooling, but rather as a possible contributor. We have also listed changes tectonics and topography as possible mechanisms proposed to drive the E/O climate shift in the Introduction section.

L.135-139: This sentence needs to be revised as its meaning is hard to understand.

Done. It now reads (Lines 105-107): "We present newly generated mass accumulation rate (MAR) data for both groups, based on the same sediment samples used in recent biological barium (bio-Ba) reconstructions (Rodrigues de Faria et al., 2024)."

165-175: Sample preparation and accumulation rates: For radiolarians, and considering their weight (particularly those from the Eocene), is a sample weight of less than 1g suitable for conducting quantitative analysis? A too small weight could increase the margin of error in measurements, and it might not provide enough material for robust statistical analysis. Then, counting only a small area of the slide to get absolute abundances is theoretically good, but it relies on the assumption that your slide is perfectly homogeneous. How homogeneous are your slides? I have tried many ways to get absolute abundances, but I found that Itaki et al.'s (2018 IODP 346 Data Report) method, which involves mounting specific Q-slides using micropipette and scanning the whole Q-slide, to be the most accurate at that time.

Thank you for rising these important points. In our study, most samples used for radiolarian analysis were around and above 1 g, which we consider sufficient for robust quantitative analysis. In the cases were sample weights were below this (especially ODP 1090 samples), we relied on the long-standing experience of our group with radiolarian preparation and counting, where similar sample sizes have consistently yielded reliable results. Also, as part of a parallel project focusing on radiolarian diversity, additional slides were prepared from the same samples, and the observed abundances were consistent with the measurements reported here.

Regarding homogeneity, our slides were prepared following the method of Lazarus (1994)*, which ensures a representative distribution of microfossils across the slide.

*Lazarus, D.: An improved cover-slip holder for preparing microslides of randomly distributed particles, Journal of Sedimentary Research, 64, 686–0, https://doi.org/10.2110/jsr.64.686, 1994.

252-257: to not fit with Ba proxy data also happened in the Japan Sea for the late Miocene, where Matsuzaki et al. (2022, Scientific Reports) data did not really fit with Zhai et al.'s 2021 Paleoceanography Ba data.

Thank you very much for this valuable observation. We do point out the gap between bio-Ba and biogenic silica in the paper, but we did not go into detail about the specific factors behind this lag. Our aim was to keep paper focused and to the point, without moving into side discussions beyond the main scope.

L.289: What do you mean by "mode of opal productivity pattern"? Please explain.

We have now revised the text and stated what we mean by it. It now reads (Lines 275-276): "Although the timing and **mode of opal productivity (that is, the relative contribution of diatoms and radiolarians, the underlying diatom diversity, and whether the flux is pulsed or gradual)** differ among sites, …"

L.296-297: Please explain what happens to radiolarians in more detail.

We have elaborated on the radiolarian discussion in the first two sections of the Discussion (Sections 4.1 and 4.2). In addition, we revised the figure presenting radiolarian results (Fig. 4) and added new supplementary figures showing changes on radiolarian trends (Fig. S2 and S7), including both latitudinal patterns and diatom-radiolarian dominance trends, which are also referenced in the text.

L.305-312: Can this lag be associated with age model uncertainty? Because you did not provide information about that, it is hard to judge and thus I ask you to add this explanation. Actually, I assume there are low sedimentation rates, and usually the Eocene has a ±0.5 Ma error for each biostratigraphic datum used which can easily give a 1 Ma offset...

Thank you for this comment. We agree that age model uncertainties can often lead to apparent lags in paleoceanographic records. However, in this case, our diatom MARs are based on exactly the same samples, age model, and age-depth control points used in the bio-Ba study by Rodrigues de Faria et al. (2024). Therefore, the observed offset in timing between the bio-Ba productivity peaks (~37 Ma) and diatom MAR peaks (36–35.5 Ma) at Antarctic sites (ODP 689 and 748) cannot be attributed to age model differences. We have now clarified this point in the revised manuscript. In addition, detailed age model information is now provided in **Supplementary Text 1**.

The highlighted text now reads (Lines 285-292): "Bio-Ba records from ODP Sites 1090, 689,

and 748 (Rodrigues de Faria et al., 2024) document a ~37 Ma export productivity peak that is synchronous with the diatom MAR rise at ODP Site 1090 but leads peaks at Antarctic-adjacent sites (ODP 689, 748) by ~0.5–1 Myr (Fig. 2e). **This offset does not reflect differences in age models, as identical samples and age models were used in both datasets,** and likely reflects regional environmental controls during middle-to-late Eocene, including latitudinal differences in sea-surface temperature (e.g., Douglas et al., 2014; Sauermilch et al., 2021), variations in nutrient distribution, and circulation patterns influenced by still-shallow SO gateways (e.g, Sauermilch et al., 2021; Rodrigues de Faria et al., 2024)."

L.346: I don't see any part of the discussion addressing radiolarian contribution except here as evidence. As you discuss radiolarians later, I would suggest focusing only on diatoms here.

We have revised the Discussion section so that radiolarian results are now discussed together with diatom records in the Sections 4.1 and 4.2, and the separate radiolarian subsection has been deleted as suggested.

---

## Author Response (AR2)

**Dear Editor Dr. Antje Völker,**

We are genuinely grateful for your detailed comments and careful checks for our manuscript. We have addressed all points and made the necessary revisions.

Below, we provide our responses. Please note that for minor wording edits and typographical corrections, we do not list individual responses. All such changes have been applied, and we have carefully rechecked the entire text for consistency and remaining issues.

Once again, we thank you for your time and thoughtful guidance.

All the best,

Volkan and co-authors

Line 55: I find underlying an awkward phrasing in this context. I suggest to use something like "resulting in" or "leading to".

Fixed. It now reads: "… the broader mechanistic mosaic **leading to** the E/O transition and Antarctic glaciation."

Line 65: A potential additional reference (since there are so few around related to diatoms) could be: Abrantes, F., Cermeno, P., Lopes, C., Romero, O., Matos, L., Van Iperen, J., Rufino, M., Magalhães, V., 2016. Diatoms Si uptake capacity drives carbon export in coastal upwelling systems. Biogeosciences 13, 4099-4109, doi: 10.5194/bg-13-4099-2016.

We thank you for this recommendation. We have now added it accordingly. We also note that this important work is cited in Section 4.2.

Line 85: latest would probably be better wording or "ending of the"

Fixed.

Line 88: If meant related to the stratigraphic subunits, both Mid(dle) and Late should be capitalized.

We have corrected all highlighted instances and carefully checked the manuscript to ensure consistent syntax of stratigraphic terms throughout the text.

Line 111: I would say youeither analyzed the …abundanceor produced the …. abundance data

We fixed the highlighted part, it now reads: "We **generated** diatom and radiolarian abundance data from samples"

Line 123: I believe with the shortening of the introduction EOT as acronym has not been defined, yet.For easier reading E/O transition as you frequently use in the subsequent text might be the better solution and in that case you don't need to define another acronym.

Thank you. We have followed your recommendation and now use "E/O transition" consistently through the text.

Line 123: If you defined EOT as abbreviation you need to use it in the subsequent text like also two line below.

Please see our comment above.

Line 184: Check the syntax here; this incomplete sentence refer to what? number of samples for the MAR average and SD at Site 1090?

We corrected the unclear syntax and the corresponding part now reads:

"Site 1090 had an average diatom MAR of $1.26 \times 10^7$ frustules cm$^{-2}$ kyr$^{-1}$ (standard deviation (std. dev.) = $9.48 \times 10^6$; range: $5.88 \times 10^5$ to $3.26 \times 10^7$; total number of samples, N = 15).

Lines 222-229: This whole paragraph is discussion and not results! And you should provide a few details on the associated environmental conditions.

We agree that this paragraph was interpretive and overlapped with our later discussion (Sections 4.1 and 4.2) of these intervals. To avoid redundancy, we have removed it from the results section.

Line 285: May be insert here "plankton" to reiterate that (so far) your are discussing surface ocean data.

Added.

Line 289: May be replace peak with maximum, so that you don't always use the same wording

Replaced.

Line 299: not epsilon Nd data? if yes, please correct wording used here.

Fixed.

Line 308: verify that you defined this acronym *(SST)* before. I don't remember reading it.

Fixed.

Line 331: Since you used this phrase *(proto-ACC)* already above you should provide this information when you use it the first time.

Done. We have now introduced the full term "proto–Antarctic Circumpolar Current (proto-ACC)" at its first mention in the manuscript, Line 306.

Line 324: What do you mean by increasing circulation? Increased transport? Increased depth penetration of the proto-ACC (thicker ACC layer? wider ACC?)?

Thank you very much for pointing this out. The original text is indeed vague. We revised the sentence to clarify what aspect of circulation we mean. It now reads:

"…are interpreted here as a response to **the intensification of** SO circulation, **likely reflecting stronger eastward flow and enhanced vertical exchange within the developing proto-ACC system,** and **the** associated **increase** in nutrient distribution and upwelling."

Line 360: if you use the phrase/ acronym only a few times in the manuscript I recommend to minimize the acronyms introduced in the text. Fewer acronyms to remember, especially those readers might not be so familiar with, makes following the text and arguments easier.

Thank you for this helpful note. We removed the acronyms for the Tasmanian Gateway and the Priabonian Oxygen Minimum.

Line 360: actually a shift from zooplankton to phytoplankton; so, a shift in primary productivity

Thank you for comment. We revised this section to clarify our interpretation. We now specify that the transition to a diatom-dominated regime around 33.5 Ma reflects a change in primary productivity and a more direct coupling between silica utilization and organic carbon export, rather than just an increase in opal deposition. This example highlights, once again, the importance of distinguishing the underlying sources of bulk opal records, which is one of the main motivation of our research.

It now reads: "This shift occurs while overall opal flux remains low in the Antarctic-adjacent sites, pointing to **a change in the biological source of silica, with production increasingly driven by diatoms. Such a transition implies a more direct coupling between silica utilization and organic carbon export, even under relatively low total fluxes."**

Line 370: should be encapsulating but I also find the word choice awkward/incorrect. may be use encompassing

Done.

Line 405: Please provide some details, e.g., species richness increases from X to Y or H (Shannon) index increases from X to Y; a reader should not be forced to read your other manuscript in order to follow the argumentation in this manuscript..

Thank you so much for this point. In our recent work (Özen et al., subm.; https://doi.org/10.31223/X50N1B), we documented that SO diatom assemblages were far more diverse than previously recognized. Diversity increased notably across the Middle-to-Late Eocene transition and the Eocene/Oligocene boundary but was not uniform

throughout the entire Eocene/Oligocene transition. Overall, diatom assemblages appear at least five times more diverse than previously reported, revealing a highly dynamic and compositionally variable community structure. This observation aligns better with our argument in this paragraph that higher diversity corresponds to a broader range of ecological strategies (e.g, niche partitioning) and thus functional traits (e.g., Tréguer et al., 2018*), leading to more efficient nutrient utilization, a key factor influencing biological pump. We have revised the text accordingly, which now reads:

"**Recent diversity reconstructions** (Özen et al., subm.) **show that Late Eocene–Early Oligocene SO diatom assemblages were at least five times more diverse than previously documented, revealing a pronounced increase in species richness and a highly dynamic community composition throughout this interval. Such diversity** is expected to be positively associated with a broader range of functional traits within the community (Tréguer et al., 2018), and **therefore with more efficient** nutrient utilization, which is one of the operating terms **of** the biological carbon pump (Farmer et al., 2021)."

*Tréguer, P., Bowler, C., Moriceau, B., Dutkiewicz, S., Gehlen, M., Aumont, O., Bittner, L., Dugdale, R., Finkel, Z., Iudicone, D., Jahn, O., Guidi, L., Lasbleiz, M., Leblanc, K., Levy, M., and Pondaven, P.: Influence of diatom diversity on the ocean biological carbon pump, Nature Geosci, 11, 27–37, https://doi.org/10.1038/s41561-017-0028-x, 2018.

Line 427: May be look also at: Rigual-Hernández, A.S., Pilskaln, C.H., Cortina, A., Abrantes, F., Armand, L.K., 2019. Diatom species fluxes in the seasonally ice-covered Antarctic Zone: New data from offshore Prydz Bay and comparison with other regions from the eastern Antarctic and western Pacific sectors of the Southern Ocean. Deep Sea Research Part II: Topical Studies in Oceanography 161, 92-104, https://doi.org/10.1016/j.dsr2.2018.06.005.

We thank you for suggesting this reference. We have now incorporated Rigual-Hernández et al., (2019) into the text to complement the modern ocean perspective on the relationship between diatom abundance diversity.

Line 462: Please give some specifics; see comment above; shifts from what type of community to what other type (or dominant species).

We revised the text to clarify the type of community shifts we are exactly referring to. It now reads: "Importantly, these productivity events coincide with **changes in dominant species composition and community structure** (Özen et al., subm.), **which** may have tuned the efficiency of the biological carbon pump (Tréguer et al., 2018)."